# FoGGAN: Generating Realistic Parkinson’s Disease Freezing of Gait Data Using GANs

**DOI:** 10.3390/s23198158

**Published:** 2023-09-28

**Authors:** Nikolaos Peppes, Panagiotis Tsakanikas, Emmanouil Daskalakis, Theodoros Alexakis, Evgenia Adamopoulou, Konstantinos Demestichas

**Affiliations:** 1Institute of Communication and Computer Systems, National Technical University of Athens, 15773 Athens, Greece; ptsakanikas@cn.ntua.gr (P.T.); edaskalakis@cn.ntua.gr (E.D.); talexakis@cn.ntua.gr (T.A.); eadam@cn.ntua.gr (E.A.); 2Department of Agricultural Economics and Rural Development, Agricultural University of Athens, 11855 Athens, Greece; cdemest@aua.gr

**Keywords:** Parkinson’s Disease, freezing of gait (FoG), GAN, DNN, artificial intelligence (AI)

## Abstract

Data scarcity in the healthcare domain is a major drawback for most state-of-the-art technologies engaging artificial intelligence. The unavailability of quality data due to both the difficulty to gather and label them as well as due to their sensitive nature create a breeding ground for data augmentation solutions. Parkinson’s Disease (PD) which can have a wide range of symptoms including motor impairments consists of a very challenging case for quality data acquisition. Generative Adversarial Networks (GANs) can help alleviate such data availability issues. In this light, this study focuses on a data augmentation solution engaging Generative Adversarial Networks (GANs) using a freezing of gait (FoG) symptom dataset as input. The data generated by the so-called FoGGAN architecture presented in this study are almost identical to the original as concluded by a variety of similarity metrics. This highlights the significance of such solutions as they can provide credible synthetically generated data which can be utilized as training dataset inputs to AI applications. Additionally, a DNN classifier’s performance is evaluated using three different evaluation datasets and the accuracy results were quite encouraging, highlighting that the FOGGAN solution could lead to the alleviation of the data shortage matter.

## 1. Introduction

PD according to the World Health Organization is a degenerative condition of the brain, associated with motor symptoms (e.g., tremor, imbalance, slow movement, and FoG) as well as non-motor symptoms (e.g., insomnia, cognitive impairment, pain, and sensory disturbances) [1]. The symptoms usually emerge slowly, and as the disease worsens, non-motor symptoms become more apparent [2]. Early symptoms are tremors, rigidity, slowness of movement, and walking difficulties [3]. Issues during the disease progression include cognition, mood, sleep disorders (also as prodromal signs), and various sensory systems deficits [4]. FoG is a unique and higher gait disorder in advanced PD patients defined as a “brief episodic absence or marked reduction of forwarding progression of the feet despite the intention to walk” [5]. The symptom lasts a couple of seconds or more and poses many difficulties to clinicians in understanding its exact mechanisms and finding a proper treatment [6,7]. About half of the people with PD exhibit freezing of gait episodes, where the most common and initial symptoms are trembling in place with no motion, shuffling, or hastening, and total akinesia with tremor and hastening. FoG occurs during the gait initiation and turning but manifests in constraints like walking through a narrow path, doorways, dual tasking, etc., which are different for each individual.

The number of PD cases has been increasing in recent decades, at a faster pace as compared to other neurological diseases. In the United States, nearly 90,000 people are diagnosed with PD every year and according to the Parkinson’s Foundation, there will be 1.2 million people with PD in the US by 2030 [8] whilst worldwide it is calculated that there are around 10 million people having this condition [9]. This raises concerns among epidemiologists and has attracted the interest of the academic, research and health communities. The reasons why PD cases increase year by year include increased stress, lack of physical exercise, aging population as well as better medical treatment which leads to prolonged life duration of people as well as of PD patients. The patients can continue their life without any serious effects when early detection can lead to the right treatment and timely interventions. Thus, it is of utmost importance to gather data for developing techniques and tools that detect and fight the PD early.

FoG is recognized as one of the most critical debilitating motor symptoms of advanced PD, presents a higher rate of occurrence in aged people while, as elaborated previously, its episodes are random in time and subjective to each person at that occasion, and under those circumstances it manifests. Thus, the inherent difficulties and the random nature of FoG manifestation, in tandem with the need for an experienced clinician’s presence (while data acquisition occurs) to verify and annotate the data (time point of occurrence), exhibit the limitations in collecting large volumes of FoG-related data. It is apparent that data augmentation via synthetic data creation for prediction/classification purposes is more than critical towards robust and generic model development. Such tools, given the technological evolution, can be provided by computer science, i.e., AI methods. The main hindrance to such technologies is the limited availability of data in order to be sufficiently robust and efficient.

The availability of data in the healthcare domain is crucial and, in many cases, due to many reasons such as privacy legislation, there is difficulty in data gathering, with data being scarce, unstructured or of low quality. Especially in PD, another limitation concerns the patients who cannot provide daily or periodically unbiased and exact data in a systematic way due to motor impairment and other symptoms coupled with the frequency the patients visit and report to their doctors. Also, the wide spectrum of data that can be collected for PD patients imposes another difficulty for computer scientists to develop tools to detect PD in early stages. The heterogeneity and the scarcity of PD data are a major concern for state-of-the-art technologies as they can hinder such “smart” solutions due to inefficient training. This leads to the necessity of alternative ways for data augmentation by engaging state-of-the-art technology such as GANs.

Ian Goodfellow et al. [10], back in 2014, proposed the term and a framework called Generative Adversarial Network (GAN). GANs have the ability to generate almost identical data records as those provided as input to the generator merged with random noise. Also, they consist of an alternative technique for developing generative models and architectures. GANs have proved quite effective and useful for applications that require data augmentation as is the case for the one examined in this study. In this light, this very study aims to present the usefulness of GANs to the data augmentation of FoG data samples for PD patients. In the methodology presented, there is a thorough description of the parameterization as well as the architecture of different GAN implementations in order to evaluate and compare the synthetically generated data. Through various indicators, the quality of the data generated is assessed in order to conclude the similarity of them with the original provided data. This solution aims to lift the limitations imposed by lack of data or unstructured and low-quality data in the domain of the PD. Researchers and academics having available such solutions in their quiver can make significant progress in terms of AI solutions in domains where data availability is a major issue.

The shortage of data in the health domain, due to the sensitive and private nature of healthcare data which is accompanied with strict ethical and legal regulations governing their acquisition and usage is, in general, a well-known bottleneck for efficient model training, because it is hard to acquire. This can result in limited sample sizes, restricting dataset diversity and representativeness. The lack of quality data significantly impacts the development of AI technologies in multiple ways: (i) affecting the model’s performance, (ii) introducing biases leading to inaccurate and unfair predictions, (iii) raising ethical concerns regarding patient privacy, (iv) causing delays in AI model development, and (v) impacting validation compliance, as adequate data are vital for model efficacy. Thus, the motivation and the contribution of this work is the development of a novel yet efficient methodology for synthetic data creation. In this work, we present a novel approach for realistic synthetic data creation with the exact characteristic properties of the data fed in the FoGGAN model, that its application can lead to data augmentation and mitigation of bias and unbalances by creating data using the under-represented origin of the original data. This highlights the potential of GANs in mitigating data scarcity issues by generating data that preserve essential and similar-looking statistical and structural characteristics of the limited original input dataset.

The remainder of the paper is organized as follows: Section 2 presents related works, focusing on the domain of PD and healthcare data augmentation. Section 3 describes the dataset used and the methodology designed and developed, whilst Section 4 elaborates on the produced results. Finally, Section 5 concludes the paper.

## 2. Related Works

There are many examples in the current scientific literature regarding the use of GANs for the generation of synthetic data (e.g., tabular data, image data, and audio data) related to the health domain. In [11], Hargreaves and Heng examined the potential of using GANs to generate synthetic tabular diabetes data. The authors built two models for the classification of patients as having diabetes or not, where the first utilized real data only, while the second made use of a combination of real and GAN-based synthetic data. The second model achieved a classification accuracy of 87.0%, yielding an increase of 8.3% as compared to the first model. The synthetic data were very similar to the original dataset and were found capable of replacing real data in research applications. Choi et al. [12] combined GANs and Autoencoders to generate the medical GAN (medGAN) model which was capable of producing realistic synthetic patient records. The model could handle both binary variables and high-dimensional discrete variables. Aiming to increase the learning efficiency and to solve mode collapse, the authors also proposed a minibatch averaging method. Experimental evaluation of the synthetic data showed comparable results to models using real data only, while also helping to protect privacy, presenting a very small risk of identity/attribute disclosure. Aiming to generate more accurate synthetic data with regard to both discrete and continuous variables, Bowaly et al. [13] proposed two variations of medGAN. The first model was called medical Wasserstein GAN (medWGAN) and integrated the Wasserstein GAN with the Gradient Penalty [14] model, while the second was called the medical boundary-seeking GAN (medBGAN) and integrated the Boundary-seeking GAN [15] model. In both models, the generator and discriminator consisted of feed-forward neural networks. The proposed alterations were found to outperform medGAN in all test scenarios, utilizing the MIMIC-III [16] and Taiwanese National Health Insurance Research Database [17].

Yang et al. [18] proposed the so-called Grouped Correlational GAN (GcGAN) model for generating realistic synthetic Electronic Health Records (EHRs). The model took into consideration the meaning of diverse variables it contained as well as the correlations among them. The authors utilized spectral normalization on the discriminator as well as batch normalization on the generator. In terms of the percentage of the qualified synthetic data, it reached 95.21% during experimental evaluation, outperforming other state-of-the art approaches such as medGAN, wGAN [19], ehrGAN [20], and CorrGAN [21]. Yoon et al. [22] proposed the (Anonymization through Data Synthesis GAN) ADS-GAN model for generating synthetic EHRs, by closely approximating the joint distributions of the used variables. Setting the patient’s privacy as a priority, the authors highlighted that the model minimized the possibility of identifying a patient based on the data which were present in the original dataset. The model was also very reliable in joint distribution and consistently outperformed other contemporary approaches such as the PATE-GAN [23], DP-GAN [24], and MedGAN. Wang et al. [25] proposed the so-called Sequentially Coupled Generative Adversarial Network (SC-GAN) for generating synthetic data relevant to both the patient state and the medication dosage. The model made use of two coupled generators (LSTM with two layers), the first about the patient state and the second about the medication dosage. The authors underlined that the patient state and medication dosage were strongly interrelated. SC-GAN was tested experimentally, outperforming other models (e.g., SeqGAN [26], and C-RNN-GAN [27]) in the medication dosage recommendation task with regard to the precision and AUROC metrics. In [28], Beaulieu-Jones et al. demonstrated the Auxilliary Classifier GAN (AC-GAN) to specify the treatment class of patients as standard or intensive. The generator in the specific model made use of the noise vector and actually knew the type of treatment class it needed to create. Differential privacy was also applied during the generation of synthetic patient data, thus helping to reduce the chance of identifying a patient based on the original data. The model was tested experimentally, proving that it can help perform hypothesis-generating analyses, with limited original trial data.

GANs are also particularly useful for augmenting time series data as well as health-related signals. Esteban et al. [29] proposed the so-called Recurrent Conditional GAN (RCGAN) which aimed at generating real-valued high-dimensional time series and focused on medical data. The generator and the discriminator encompassed Recurrent Neural Networks (RNNs) which were conditioned on auxiliary information. The synthetic data which were produced included time series and associated labels. The results generated by the synthetic data related to Internal Care Unit (ICU) patients were found to be comparable to those produced based on real data only, reaching 0.96 with regard to the AUROC metric as compared to 0.9908.

Kiyasseh et al. [30] proposed the so-called PlethAugment model encompassing three Conditional GAN [31] models with an adapted diversity term. Aiming to improve the classification performance, PlethAugment focused on producing pathological photo-plethysmogram (PPG) signals. With regard to the AUROC metric, the use of the generated synthetic dataset yielded a 29% increase as compared to the original class-balanced datasets. Brophy et al. [32] proposed a GAN-based model called Multivariate GAN (MV-GAN) for generating realistic multichannel electrocardiogram (ECG) signals. By utilizing minibatch discrimination (MBD) in the GAN architecture, the authors avoided the mode collapse problem and could generate multivariate time series. Experimental testing indicated that the synthetic datasets generated were structurally similar to the original datasets with satisfactory diversity among the different samples, while also ensuring protection of the patients’ privacy. Hazra and Byun [33] proposed a GAN-based model which had the main goal of automating and improving medical diagnosis as well as of enriching the training of medical students by utilizing realistic data. The so-called SynSigGAN model made use of Bidirectional Long Short Term Memory networks (BiLSTM) for the generator network and (Convolutional Neural Networks (CNNs) for the discriminator. SynSigGAN was used for the generation of ECG, electroencephalogram (EEG) as well as of electromyography (EMG) and photoplethysmography (PPG) signals. Experimental testing of the proposed model indicated its potential in producing realistic results with high correlation between the original and the synthetic signals. The model also outperformed other contemporary models (e.g., LSTM-AE [34], BiLSTM-MLP [35], and RNN-AE GAN [36]) in terms of Root Mean Square Error (RMSE) and Mean Absolute Error (MAE), achieving the best results (0.25 and 0.36, respectively).

Privacy is a major concern in applications encompassing data augmentation with GANs. Torfi et al. [37] proposed a framework for the synthetic generation of data, making use of the Rényi differential privacy. The authors utilized convolutional autoencoders and Convolutional GANs (CGANs) and were capable of capturing feature correlations and temporal information in the original datasets. Experimental testing of the framework highlighted its capability in generating realistic synthetic data. The framework was also found to outperform other state-of-the art approaches (e.g., MedGAN, TableGAN [38], GAN, and PATE-GAN) in terms of Area Under the Precision-Recall Curve (AUPRC) metric, reaching 0.93. Chin-Cheong et al. [39] proposed two WGAN variants for the generation of realistic heterogeneous EHRs. The first variant demonstrated quite satisfying results regarding data fidelity and data utility. More specifically, the AUROC and AUPRC metrics were comparable to the use of real data, reaching 0.7536 and 0.7747, respectively, for data utility as compared to 0.8003 and 0.8245 for the real data. The second variant also applied a differential privacy model, ensured better privacy, but had worse results (0.6427 AUROC and 0.6776 AUPRC) which, however, were still usable for ML tasks.

There are many GAN-based applications used specifically in the domain of PD diagnosis and treatment. Kaur et al. [40] proposed a model which combined GANs and Deep Neural Networks (DNNs). Early detection of the PD by voice analysis (e.g., analysis of voice strength, articulation rate, long pauses, and pitch rate) and classification. Initially, GANs were used to expand the original training dataset by generating synthetic data. These data were then used for PD classification. Experimental testing of the model highlighted that data augmentation through GANs improved the model’s accuracy and specificity as compared to the use of the original data without data augmentation. More specifically, an 88% accuracy and 87.14% specificity were achieved as compared to 84.67% and 83.76%, respectively, when no data augmentation was used.

Voiceprints used for distinguishing PD patients and healthy individuals were utilized by Xu et al. [41]. More specifically, the authors proposed a GAN-based model, which was combined with different models and classification models, helping train the aforementioned models even when very limited data were available. The so-called Spectrogram Deep Convolutional GAN (S-DCGAN) was capable of producing high-resolution spectrograms by means of increasing the number of network layers as well as by implementing a spectral normalization method and a feature-matching method. Experimental testing, using a ResNet50 model [42], achieved a high accuracy of 91.25% in voiceprint classification and a 92.5% specificity. The authors also highlighted that the data augmentation also played a significant role in these results as the application of ResNet without the use of GANs for data augmentation resulted in much worse results (75.5% accuracy and 72.5% specificity).

Contrary to most methodologies for classifying healthy individuals and patients with dementia which focuses on one type of dementia only, Noella and Priyadarshini [43] proposed a system which can help in the diagnosis of different types of dementia. More specifically, Brain Fluorodeoxyglucose Positron Emission Tomography (FDG-PET) scans were utilized to diagnose PD, Alzheimer’s disease and Frontotemporal Dementia. GANs were used for the generation of synthetic (Neuroimaging in Frontotemporal Dementia NIFD) samples to solve uniform distribution problems in the images used for training the Deep Convolutional Neural Network (DCNN) classification model. The proposed system was tested experimentally, yielding 97.7% accuracy, 97% specificity, and 97% sensitivity.

In [44], Zanini and Colombini proposed two methodologies for augmenting the EMG signals of PD patients. The first methodology was based on Deep Convolutional GANS (DCGANs). In this case, the generator simulated the EMG tremor pattern of each patient. The discriminator of this methodology was also used on the second methodology for augmenting EMG signals, making use of neural style transfer. Experimental testing of the methodologies indicated their capability of adapting to different tremor frequencies and amplitudes of patients. The methodologies could also help extend tremor patterns to diverse movement protocols and scenarios.

Kaur et al. [45] demonstrated an approach for classifying Magnetic Resonance (MR) images as belonging to PD patients or healthy people. The approach was based on DCNNs for the classification task, while a GAN model was used for data augmentation, addressing the issue of the limited size of the available training dataset. The authors applied preprocessing of the MR images and transfer learning was implemented to the pre-trained Alex-Net architecture. The last layers of the model were replaced with new categories of images, tailored to the needs of PD classification. Experimental testing of the authors’ approach yielded a classification accuracy of 89.23%. The analysis of digital drawing tests could help in the diagnosis of PD as well as in the investigation of graphomotor impairment in PD patients [46]. Towards this direction, Dzotsenidze et al. [47] proposed a framework for conducting PD diagnostics based on digital drawings, utilizing CNNs for classification purposes combined with GANs for data augmentation. More specifically, four different GAN architectures (i.e., ProjectedGAN [48], StyleGAN3 [49], StyleGAN2-ADA [50], and StyleGAN2-ADA + LeCam [51]) were used and evaluated for generating synthetic digital drawing tests. Regarding the sensitivity metric, ProjectedGAN reached 96.6% in some test scenarios, and the authors highlighted that the use of GANs could help face data scarcity regarding digital drawing tests and contribute to better decision making for doctors.

GANs can also be used in applications relevant to the FoG symptom. Ramesh and Bilal [52] presented a model utilizing GANs and CNNs for predicting the Postural Instability and Gait Disorder (PIGD) score of PD patients wearing a single inertial sensor. The specific score was calculated, utilizing different scores related to the posture and the gait of the patient (e.g., FoG, posture, and gait). The model was also able to classify the ON/OFF states of a PD patient, with the ON state referring to when a patient has been treated with a dopamine precursor drug and the OFF state referring to the same patient when the specific drug has started to wear-off, followed by a worsening of motor symptoms [53]. The authors used data from different clinics for the training and testing of their model. The experimental results indicated that the CNN model using GANs outperformed the CNN model where no GANs were utilized, yielding an accuracy improvement of up to 22% in determining the ON/OFF states and even outperformed clinicians in determining the ON/OFF states making use of the PIGD scores. Yu et al. [54] demonstrated an approach utilizing GANs and the Hidden Markov Model (HMM) [55] for classifying whether or not to activate devices which protect patients with chronic diseases from falling. The authors’ approach alleviated many problems present in models for chronic conditions such as relying on manual feature engineering and omitting temporal dependencies. The so-called HMM-GAN model was capable of capturing independent and sequential data from sensors which followed diverse distributions. Experimental testing of the model under both supervised and semi-supervised settings showcased increased accuracy in predicting if the protective equipment should be triggered or not. Specifically, the model’s accuracy reached 93.07% in the supervised mode and 94.82% in the semi-supervised mode. The authors noted that their approach can also be used for recognizing FoG symptoms.

## 3. Data Feature Selection and Generation Using GANs

### 3.1. Dataset

The ‘data_daphnet_combined’ dataset [56] was based on the Daphnet Freezing of Gait dataset [57] which was devised to benchmark automatic methods to recognize gait freeze from wearable acceleration sensors placed on the legs and hip. The dataset can be used in research and the evaluation of machine learning models for PD detection. It provides a realistic and representative sample of sensory records of PD patients, making it a valuable resource for researchers and practitioners in the field of health. The ‘data_daphnet_combined’ dataset had twelve columns which contained nine different attributes as well as a time, an annotation, and a filename column. The Daphnet Freezing of Gait dataset captures were collected in the lab with emphasis on generating many freeze events. Users performed their kinds of tasks: straight line walking, walking with numerous turns, and finally a more realistic activity of daily living (ADL) task, where users went into different rooms while fetching coffee, opening doors, etc.

The dataset contained a total of 1.92 million records. Each record contains a value (real number) for each of the nine attributes collected. Also, for each record, there is the annotation column which can have a 0, 1, or 2 value. These annotations mean the following:0: not part of the experiment. For instance, the sensors were installed on the user or the user was performing activities unrelated to the experimental protocol, such as debriefing;1: experiment, no freeze (can be any of stand, walk, or turn);2: freeze.

To refine the dataset and optimize the generation processes, we executed targeted preprocessing procedures. First, we excluded data instances categorized under class 0. To simplify the classification task, that will be detailed in Section 4, we redeclared class 2, originally denoted as “freeze” events, as class 0. Meanwhile, class 1 remained unprocessed throughout this process.

Thus, each data record in the dataset included detailed information about time, the values from the sensors placed to ankles, upper legs and the trunks as well as the annotation and the data file containing the record. Table 1 presents the data type of each of the 12 features included in the initial dataset used.

### 3.2. FoGGAN Architecture

Generative Adversarial Networks (GANs) are a category of algorithms, which encompass a dual neural network framework characterized by adversarial competition, hence the term “adversarial”. This architectural solution comprises two distinct neural networks, specifically referred to as the generator and the discriminator, collaborating to generate synthetic data. In 2014, Ian Goodfellow and his colleagues introduced [10] advanced deep learning techniques aimed at generating diverse types of synthetic datasets, encompassing images, tabular data, text, videos, and music compositions, giving a strong emphasis on achieving a high degree of similarity to the original datasets.

The primary objective of the generator is the creation of top-tier synthetic data, with the specific intent of deceiving the discriminator. It takes a random noise vector as input in order to produce high-quality, similar-looking data resembling the provided content.

In contrast, the discriminator is tasked with distinguishing between real and synthetic data. The model is implemented as a sequential deep neural network, comprising dense and dropout layers, with the task of classifying input data samples as either real (original) or fake (generated). Its effectiveness in distinguishing real from fake data samples is, then, utilized to optimize and enhance the overall performance of the GAN, encompassing both the generator and the discriminator. Figure 1 illustrates the adversarial competition between the generator and the discriminator as well as the overall flow of processes in this architecture. Once the data samples are appropriately classified as either real or fake by the discriminator, it returns the corresponding feedback to the generator to readjust and improve its weights accordingly so as to continue the data-sample-generation process.

Calculating generator and discriminator losses is of paramount importance during the training processes of Generative Adversarial Networks (GANs). These loss functions serve as pivotal metrics to optimize the performance of both networks. They facilitate the adversarial learning process by quantifying the generator’s ability to deceive the discriminator and the discriminator’s capability to distinguish between real and generated data samples. As previously mentioned, and demonstrated in Figure 1, the losses establish a critical feedback loop, driving iterative improvements in both the generator and discriminator. Furthermore, they are instrumental in achieving GAN convergence, where the generator generates realistic data and the discriminator performs at chance level, ensuring the production of high-quality and comparable synthetic data in appearance.

The pivotal aspect of the GANs’ evolvement lies in the utilization of loss functions, a collection of mathematical Equations (1)–(3) guiding each network’s improvement after every training epoch. These equations are provided later in the text. The discriminator and generator possess their respective loss values. Across successive epochs, these networks learn by striving to minimize their respective loss functions.

More specifically, the generator and discriminator losses are computed independently and then integrated through a min-max game as described below by Equation (1) [10]. In this equation, *G* represents the generator, *D* represents the discriminator, and *V*(*D*, *G*) represents the value function of the min-max game. In greater detail, the process begins by establishing the generator’s data distribution, denoted as *p_g_*(*x*), which operates under the assumption that the input noise variables, *p_z_*(*z*) have been already defined. Once these noise variables are defined, a mapping to the data space is articulated as *G*(*z*; *θ_g_*), where *G* represents a differentiable function instantiated as a multilayer perceptron, characterized by parameters *θ_g_*. Simultaneously, a second multilayer perceptron, denoted as *D*(*x*; *θ_d_*), is introduced. This perceptron returns an output of a singular scalar value. Specifically, *D*(*x*) quantifies the probability that the data point x originates from the actual data distribution rather than being generated by *p_g_*. Consequently, the training procedure involves dual objectives: Firstly, the discriminator is trained to maximize the likelihood of correctly classifying both original and generated samples. Secondly, the generator is trained to minimize the negative logarithm of (1 − *D*(*G*(*z*))).

Additionally, the losses for both the generator and the discriminator can be computed independently using Equations (2) and (3). Both generator and discriminator losses will ultimately converge to a stable state as they undergo an adequate number of training epochs.
(1)$$min_{G}max_{D}V\left(D,G\right)=E_{x∼p_{g}\left(x\right)}\left[\log D\left(x\right)\right]+E_{z∼p_{z}\left(z\right)}\left[\log\left(1-D\left(G\left(z\right)\right)\right)\right]$$
(2)$$min_{G}V\left(G\right)=\nabla_{\theta_{g}}\frac{1}{m}\sum_{i=1}^{m}\log\left(1-D\left(G\left(z^{\left(i\right)}\right)\right)\right)$$
(3)$$max_{D}V\left(D\right)=\nabla_{\theta_{d}}\frac{1}{m}\sum_{i=1}^{m}\left[\log D(x^{\left(i\right)}+\log\left(1-D\left(G\left(z^{\left(i\right)}\right)\right)\right)\right]$$

In this study, we implemented the so-called Freeze of Gait GAN (FoGGAN) model architecture, designed specifically for generating one-dimensional (1D) synthetic data from the dataset previously described in Section 3.1. The implementation was carried out using TensorFlow 2.0 [58], leveraging the high-level Keras API. For the visual representation of the generator model’s architecture, please refer to Table 2. We employed the sequential API to construct a sequence object, effectively stacking the various layers of the proposed deep neural network. The generator component within the FoGGAN architecture comprised an input layer, accepting appropriately scaled random noise, followed by nine hidden layers, all activated using the ‘ReLU’ function, and culminating in an output layer. This output layer was activated by the ‘linear’ function and matched the dimension of the preprocessed dataset. Subsequently, Table 3 provided a comprehensive definition of the discriminator model, which was also structured as a straightforward sequential model featuring eleven dense layers, The first ten layers utilized the ‘ReLU’ activation function, while the output layer employed ‘sigmoid’ activation, serving to distinguish input samples as either real or fake. Furthermore, a dropout rate of 20% was applied to both the input (the visible one) layer and the two hidden layers of the discriminator model. The FoGGAN model underwent training for 500 epochs with a batch size of 50. Additionally, the learning rate for the discriminator was set to 0.001, while for the generator, it was 0.01.

## 4. Results

### 4.1. Comparison Results between Original and FoGGAN-Generated Data

Diagrams prove to be an efficacious tool for comparing and visualizing similarity scores between real and synthetic datasets generated by a GAN (FoGGAN in our study) model. These scores offer crucial insights into the quality and precision of the synthetic dataset, aiding researchers in pinpointing areas where improvements to the GAN model are needed to produce more similar-looking synthetic data. The choice of diagram type depends on the data’s inherent characteristics as well as on the specific objectives of the research.

In the context of the current study, we employed the developed FoGGAN architecture (as previously mentioned in Section 3.2) to replicate synthetic data from a genuine input dataset, specifically the ‘data_daphnet_combined’ dataset, detailed in Section 3.1. The generated dataset was meticulously compared to the real one to extract the corresponding similarity scores across the encompassed features (variables).

For this purpose, five distinct types of diagrams, outlined below, served as effective means to represent these similarity scores. Each figure presented in the subsequent section incorporated the following elements:Heatmaps depicting correlation matrices offer a valuable solution in terms of visualizing clusters and detecting dissimilarities between the real and generated datasets. These heatmaps prove especially beneficial for pinpointing patterns of similarity scores linked to distinct data features.Cumulative sum (or cumsum) diagrams provide a visual representation of the cumulative sum for both the real and generated datasets. In the context of assessing similarity scores and comparing datasets using the FoGGAN model, the cumsum diagram offered an effective way to visualize the accumulation of the similarity scores computed between the original and generated datasets gradually.Logarithmic (Log) mean and standard deviation (STD) diagrams usually serve as tools for comparing similarity scores between the original dataset and the one generated by a GAN (the ‘FoGGAN’ in our study). A Log mean diagram provided a visual representation of the average or mean similarity score between the original and generated dataset(s) for each training epoch. This depiction enabled an assessment of how the similarity score evolves over time, revealing whether the generated dataset’s similarity to the real dataset is increasing or decreasing during the training process. Conversely, a standard deviation diagram (STD) illustrated the variability in similarity scores between the real and generated datasets for each training epoch. This visualization assessed the consistency of the similarity score and identifies significant fluctuations in similarity between epochs.Principal Component Analysis (PCA) diagrams are employed as a valuable tool in comparing similarity scores between the original and generated datasets. These diagrams offer a graphical representation of how the under-examination dataset’s dimensions align and diverge. By visualizing the distribution of similarity scores through PCA, it was possible to distinguish patterns and trends in the relationship between the compared datasets, illuminating the evolution of their similarity as the GAN model underwent training.Distribution diagrams for individual features are instrumental in comparing similarity scores between the original and generated datasets, on a feature-specific level. These diagrams provided a focused view of how each feature’s distribution evolved over time during the FoGGAN model’s training procedure. By analyzing these distributions separately, we gained insights into the similarity fluctuations for each feature, aiding in a more detailed assessment of the synthetic data generation process.

Through the examination of these diagrams that illustrated and compared both the real and synthetic datasets, it became feasible to gauge the FoGGAN model’s effectiveness in producing synthetic data that closely mirror the attributes, in terms of quality, of the authentic data.

The visual representation of Figure 2 employed correlation matrices with heatmaps that illustrated the differences between various pairs of values between the original (on the left side) and the generated dataset (in the middle), alongside the actual dissimilarities of them (on the right side). After the examination of the correlation matrix, it was revealed that correlation coefficients with magnitudes between 0.16 and 0.3 indicated highly significant correlations between data variables. Conversely, coefficients with magnitudes ranging from 0.1 to 0.15 suggested high correlations, while those falling between 0.01 and 0.1 signified moderate correlations. The analysis presented in Figure 2′s Correlation Matrices highlighted an ordinary high similarity among all potential pairs of compared dataset features.

Subsequently, examining the results presented in the (sub)figures included in Figure 3, there was no significant deviation noted between the synthetic and the original dataset for each of the nine (9) features. These findings yielded valuable insights, indicating a consistent and notable level of resemblance regarding trends, patterns, and thresholds during the training process, both within the original and generated data features.

The insights depicted in Figure 4 suggested that data values closely clustered around both the mean and the standard deviation (STD). The proximity of each data point to these statistical reference points appeared minimal. Consequently, the mean absolute and STD distributions showcased in the original and generated datasets (Figure 4) exposed a significant overlap. The detected overlap strongly implied a comparable data spread for each corresponding dataset, further indicating the existence of a high possibility of statistical similarity between the compared datasets.

The insights drawn from Figure 5 emphasized the noteworthy correlation observed within the depicted Principal Component Analysis (PCA) dimensions between variables in the original and generated datasets. This substantial correlation suggested that a limited number of uncorrelated variables were present. This alignment underscored an essential degree of similarity and feature correlation between the two datasets, further reinforcing their close likeness.

In Figure 6, we concentrated on the evaluation of pair-wise variable similarity through the application of Distribution Metrics techniques, as previously described. The outcome of this search uncovered a noteworthy observation: the probability distributions for pair-wise variables in both the synthetic and original datasets exhibited a remarkable consistency, occupying the same range. This pronounced convergence underscored a substantial alignment between the synthetic and the original dataset, emphatically reaffirming the fact of the existence of their significant similarity.

### 4.2. FoG Incidents Classification Using a DNN Classifiier

The primary focus of this study lies in the evaluation of the data generated by the FoGGAN model in the ‘Daphnet’ dataset context, both detailed in Section 3. We gave strong emphasis on assessing how faithfully, in terms of quality, these synthetic data samples replicated the essence of the original dataset.

To further evaluate the effectiveness of the generated samples, we employed a complementary Deep Neural Network (DNN), a commonly used Deep Learning (DL) architecture in the realm of classification, as the additional arbitrator of the data authenticity. The evaluation processes on this task involved the examination of the generated data using the DNN model. The DNN classifier was initially trained with the original dataset, as detailed in Section 3.2. For evaluation purposes, we employed three different scenarios: firstly, the DNN classifier was used to evaluate the accuracy of an unseen (data) sample from the original dataset, subsequently with a mixed dataset containing both original and generated, unseen data samples and finally with the synthetic data generated from the FoGGAN.

It is noteworthy that the DNN model utilized for this study was not optimized in terms of accuracy and loss, as the primary focus lay in assessing the quality of data generated by GANs rather than the performance of the deep learning model itself. Table 4 provides an overview of the parameters employed for the DNN classifier used in this study. The DNN classifier was composed of multiple dense and dropout layers for regularization. It had a total of 3402 trainable parameters and its final output layer consisted of 2 units/classes, making it suitable for binary classification tasks. The parameters finally used for training the DNN model were determined through a systematic tuning process. The selected hyperparameters included a learning rate of 0.001, a batch size of 64, and an epoch value of around 250. These choices aimed to strike a balance between model training speed and stability. It is important to note that in this study, the primary focus was on generating similar-looking datasets using FoGGAN, rather than fine-tuning hyperparameters for the DNN predictive model.

Table 5 provides a summary of the training and evaluation sample sizes for three distinct datasets employed in this study. 

Table 6 illustrates valuable insights into the model’s performance by demonstrating the accuracy metrics for each of the different datasets (original, mixed, and generated) provided to the DNN during the evaluation phase. This metric provided a comprehensive view of the model’s accuracy during the classification processes with the data instances included within each of the evaluated datasets.

The extracted results of the freeze of gait (FoG) classification using the Deep Neural Network (DNN) classifier underscored some interesting trends, particularly in addressing data limitations in sensitive medical domains such as Parkinson’s Disease (PD) datasets. Specifically, when evaluating the model’s accuracy across different datasets, it was evident that the classifier achieved a high accuracy rate across the board. First, when evaluated with an unseen sample of data instances from the original dataset, the model achieved an accuracy of 90.29%. This demonstrated the classifier’s ability to effectively generalize to familiar data. Remarkably, when applied to a dataset generated using the FoGGAN, the model’s accuracy further improved to 92.09%. This highlighted the potential of GAN models in enhancing the dataset’s diversity and aiding the classifier in making more accurate predictions on unseen data. Furthermore, evaluating the model with a mixed dataset containing (unseen) data instances from both the original and the generated datasets yielded an accuracy of 90.66%. This result showcased the utility of the FoGGAN in augmenting the original dataset, thereby contributing to the classifier’s overall performance and robustness. The higher accuracy achieved when evaluated on the generated dataset alone highlights that the synthetic data created by the FoGGAN contributed essentially to the classifier’s performance. This is particularly valuable in scenarios where acquiring a large and diverse dataset can be challenging. In summary, the DNN classifier exhibits strong classification performance (even not optimized since the optimal classifier is out of the scope herein), with the highest accuracy achieved when evaluated on the generated dataset. Moreover, the extracted findings underscored the significance of data augmentation techniques like GANs architectures in enhancing the classifier’s accuracy and its potential in real-world applications, such as FoG classification.

### 4.3. Discussion on the Results

Access to healthcare data is often restricted in order to protect the patient’s privacy, thus hindering the reproducibility of existing results and limiting new research. In order to surpass this problem for robust and efficient AI model development, synthetically generated healthcare data have become one of the major tools [59]. This way, privacy is preserved, and researchers and policymakers are enabled to make decisions and use methods based on realistic data. Further, health data often include information on protected attributes like age, gender, race, etc. For various reasons (i.e., the COVID-19 pandemic has exacerbated health inequities, with certain subgroups experiencing poorer outcomes and less access to healthcare), imbalanced and/or biased data are the issues that need to be handled appropriately. Synthetic data generation is again a means to overcome those issues that otherwise lead to biased, untrusted, and irresponsible AI. Taking under consideration all the above, the FoGGAN architecture presented and the supporting evaluation results on the basis of the synthetic generated data both in terms of their similarity to the real data and on the performance of a classifier we are able to support our claim that the proposed data generation approach is appropriate and suitable for use in order to surpass the aforementioned health data limitations, and privacy and ethical issues.

The current research focuses on leveraging Generative Adversarial Models (GANs), specifically FOGGAN as introduced, to address data scarcity challenges in the healthcare domain, including the freezing of gait (FoG) dataset related to Parkinson’s Disease (PD), offering promising insights. Nonetheless, it is essential to acknowledge several limitations inherent to our study:Data Source Availability: The availability of high-quality, annotated FoG datasets remains a challenge. Gathering and labeling datasets, especially for rare medical conditions like FoG, is time-consuming and resource-intensive. This limitation hinders the scalability and widespread applicability of our approach.Clinical Validation: While promising, the high-accuracy results obtained using the FoGGAN-generated data for classification purposes need further clinical validation. Real-world clinical trials and expert assessments are necessary to validate the clinical utility of the synthetic data.Data Generalization: The effectiveness of the FoGGAN architecture in generating synthetic data relies on the quality and representativeness of the input dataset. If the initial dataset has limitations or biases, these may also be reflected in the generated data. Careful curation of the input dataset is necessary to mitigate this issue.

The current study acknowledges these limitations as part of our commitment to transparency and responsible research. Addressing these challenges is crucial for the continued development and deployment of GAN-based data augmentation solutions into the healthcare domain, ultimately contributing to optimized and improved patient care and medical research.

## 5. Conclusions

The main goal of this study was to present a GAN architecture for generating almost identical medical data for PD and specifically for FoG cases. The data which were used as input for the GAN deployed in this study were from the ‘data_daphnet_combined’ dataset. Based on this kind of data input, specifically in the tabular data format, the FoGGAN architecture has been shown to be able to generate almost identical data to the real dataset as indicated by the metrics presented in Section 4.1.

To assess the added value of the proposed data generation process using the FoGGAN model, we trained a Deep Neural Network (DNN) classifier using the original dataset as detailed in Section 3.1 and evaluated it, using various testing (evaluation) inputs. The outcomes, encompassing the accuracy metric, revealed that the DNN classifier’s performance consistently maintained its high accuracy in evaluating the provided, generated data, despite the diverse datasets provided (both the mixed and the evaluated). This is another crucial supportive indication that the generated synthetic data hold the same properties as the real ones. As observed, the analysis detailed in Section 4 revealed intriguing insights following the performance of the DNN classifier when evaluated with different datasets. The model showcased praiseworthy accuracy when tested with the original dataset. Furthermore, the evaluation of the model on the mixed dataset resulted in a strong accuracy rate as well, whereas a slight boost in accuracy was observed when the generated dataset was applied and evaluated. These findings are of practical significance, particularly in scenarios where acquiring extensive and diverse datasets poses challenges. Leveraging GAN architectures for data augmentation emerges as a promising strategy to address data limitations and enhance model performance in real-world applications.

The FoGGAN architecture presented in this work can be used as a very useful tool for data augmentation in the context of PD in many ways apart from the type of data showcased, i.e., FoG-related. It can be used in PD research but not only there, since the issue of data shortage is apparent in most neurodegenerative diseases (e.g., multiple sclerosis, and dementia), where FoGGAN can play a significant and multifaced role in terms of data augmentation. The most obvious one is the creation of additional data with the same statistical properties, inherent information, and predictive power as the data fed into the FoGGAN as we already presented. Another perspective is focused data augmentation for data that are underrepresented in the original/real dataset. For example, considering a bias in data due to the participants’ age, where the age group of 40–45 years old is small compared to older people, one can create additional data for that age group (same for sex, economic status, race, etc.). Thus, FoGGAN can also be considered a bias mitigation method. Further, in the same way, unbalanced datasets can be balanced, boosting the robustness and generalization of the classification or prediction model to be developed.

The future direction that this study aims to follow is to explore and evaluate different GAN architectures in order to conclude if the quality of the generated data is sufficient. The different GAN architectures that are planned to be studied in the future are the hybrid models which engage autoencoders and GANs. Moreover, future studies can also examine the impact of hyperparameters of a GAN model on the generated data. Another future prospect of the current study is to expand the data input to also include other medical data such as data from different symptoms or different diseases or even different data formats like images or videos. In this way, there will be a generalization of the capabilities of the presented GAN architectures in different health applications or other domains as well as of the data formats that will be used as inputs. Following the paradigm of the constant learning and data expansion, and based on the GAN solution proposed in this study, future research can also include lifelong learning techniques. In this way, there will be a continuous update process of the generated datasets leading to an adaptable and expandable solution about the data scarcity issue that is present in many domains including medical science domains.

## Figures and Tables

**Figure 1 sensors-23-08158-f001:**
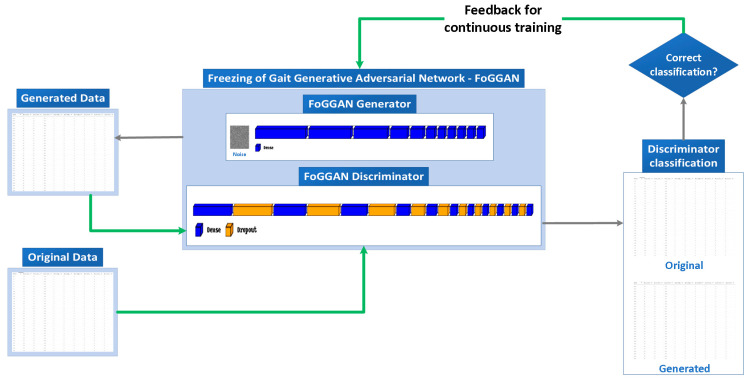
FoGGAN implementation and data flow.

**Figure 2 sensors-23-08158-f002:**
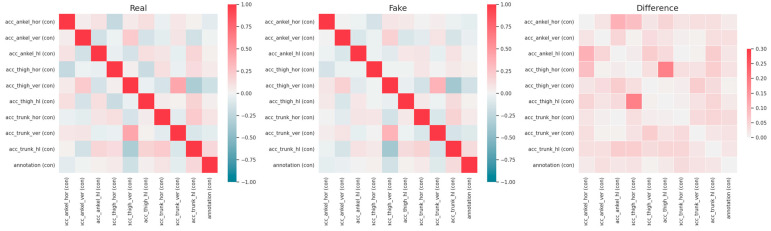
Correlation matrix for original and generated dataset.

**Figure 3 sensors-23-08158-f003:**
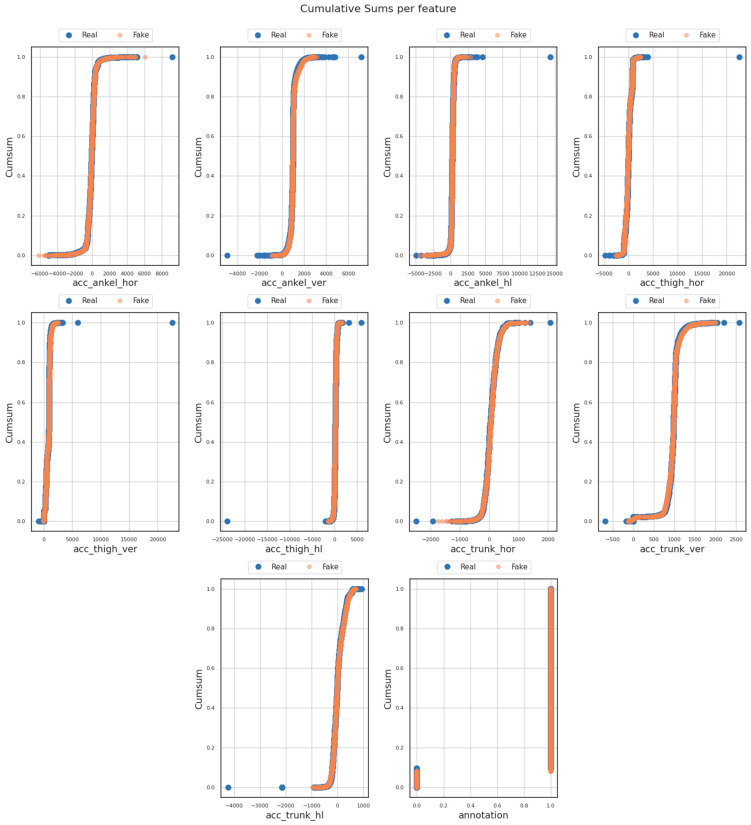
Cumulative sums per feature.

**Figure 4 sensors-23-08158-f004:**
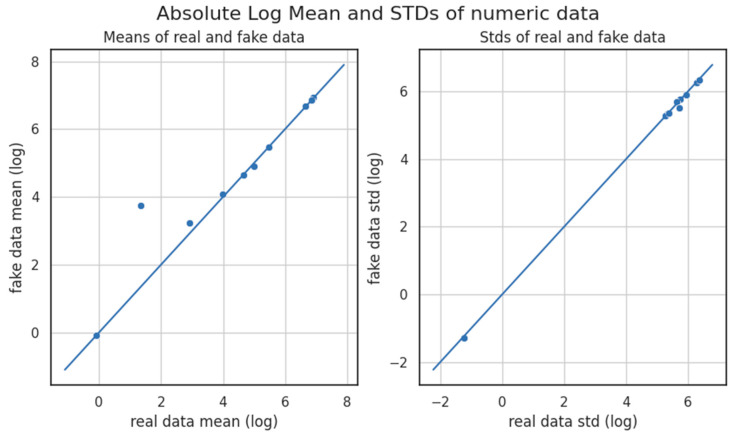
Absolute log mean and standard deviation.

**Figure 5 sensors-23-08158-f005:**
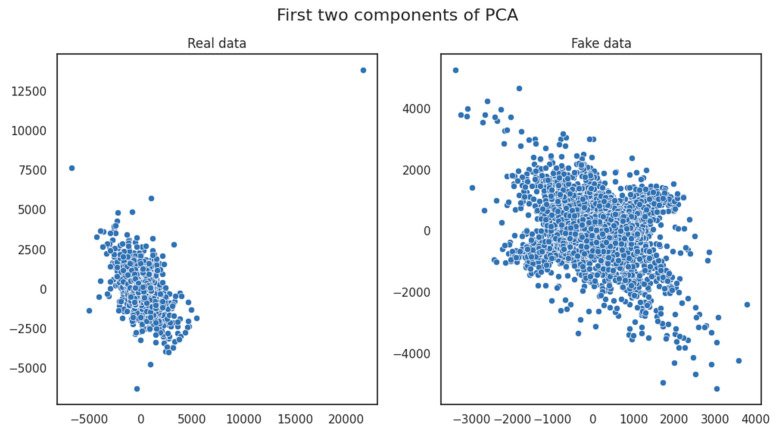
Principal Component Analysis for original and generated data.

**Figure 6 sensors-23-08158-f006:**
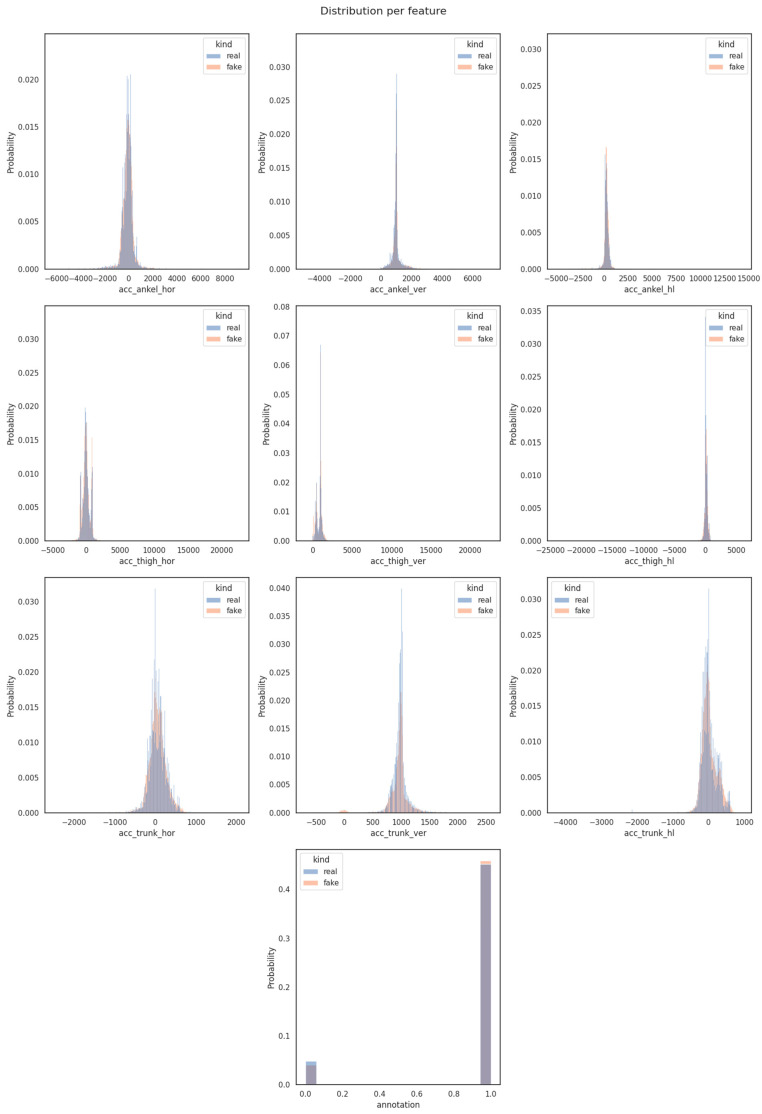
Distribution per feature.

**Table 1 sensors-23-08158-t001:** Initial dataset features and their data types.

Feature	Type	Description
acc_ankel_hor	numerical	Ankle (shank) acceleration— horizontal forward acceleration
acc_ankel_ver	numerical	Ankle (shank) acceleration— Acceleration—vertical
acc_ankel_hl	numerical	Ankle (shank) acceleration— horizontal lateral
acc_thigh_hor	numerical	Upper leg (thigh) acceleration— horizontal forward acceleration
acc_thigh_ver	numerical	Upper leg (thigh) acceleration— vertical
acc_thigh_hl	numerical	Upper leg (thigh) acceleration— horizontal lateral
acc_trunk_hor	numerical	Trunk acceleration— horizontal forward acceleration
acc_trunk_ver	numerical	Trunk acceleration— vertical
acc_trunk_hl	numerical	Trunk acceleration— horizontal lateral
annotation	numerical	0 or 1

**Table 2 sensors-23-08158-t002:** Generator model output.

Generator		
Layer (Type)	Output Shape	Number of Parameters
dense (Dense)	(None, 1536)	13,824
dense_1 (Dense)	(None, 1278)	1,964,286
dense_2 (Dense)	(None, 1024)	1,309,696
dense_3 (Dense)	(None, 512)	524,800
dense_4 (Dense)	(None, 384)	196,992
dense_5 (Dense)	(None, 256)	98,560
dense_6 (Dense)	(None, 128)	32,896
dense_7 (Dense)	(None, 64)	8256
dense_8 (Dense)	(None, 32)	2080
dense_9 (Dense)	(None, 16)	528
dense_10 (Dense)	(None, 9)	153
Total parameters	4,152,071	
Trainable parameters	4,152,071	
Non-trainable parameters	0	

**Table 3 sensors-23-08158-t003:** Discriminator model output.

Discriminator		
Layer (Type)	Output Shape	Number of Parameters
dense_11 (Dense)	(None, 1536)	13,824
dropout (Dropout)	(None, 1536)	0
dense_12 (Dense)	(None, 1278)	1,964,286
dropout_1 (Dropout)	(None, 1278)	0
dense_13 (Dense)	(None, 1024)	1,309,696
dropout_2 (Dropout)	(None, 1024)	0
dense_14 (Dense)	(None, 512)	524,800
dropout_3 (Dropout)	(None, 512)	0
dense_15 (Dense)	(None, 384)	196,992
dropout_4 (Dropout)	(None, 384)	0
dense_16 (Dense)	(None, 256)	98,560
dropout_5 (Dropout)	(None, 256)	0
dense_17 (Dense)	(None, 128)	32,896
dropout_6 (Dropout)	(None, 128)	0
dense_18 (Dense)	(None, 64)	8256
dropout_7 (Dropout)	(None, 64)	0
dense_19 (Dense)	(None, 32)	2080
dropout_8 (Dropout)	(None, 32)	0
dense_20 (Dense)	(None, 16)	528
dropout_9 (Dropout)	(None, 16)	0
dense_21 (Dense)	(None, 9)	153
Total parameters	4,152,071	
Trainable parameters	4,152,071	
Non-trainable parameters	0	

**Table 4 sensors-23-08158-t004:** DNN classifier parameters.

DL Classifier		
Layer (Type)	Output Shape	Number of Parameters
dense (Dense)	(None, 64)	640
dropout (Dropout)	(None, 64)	0
dense_1 (Dense)	(None, 32)	2080
dropout_1 (Dropout)	(None, 32)	0
dense_2 (Dense)	(None, 16)	528
dropout_2 (Dropout)	(None, 16)	0
dense_3 (Dense)	(None, 8)	136
dropout_3 (Dropout)	(None, 8)	0
dense_4 (Dense)	(None, 2)	18
Total parameters	3402	
Trainable parameters	3402	
Non-trainable parameters	0	

**Table 5 sensors-23-08158-t005:** Training, evaluation, and total samples of original, generated and mixed dataset.

Training/ Evaluation Dataset	Original	Generated	Mixed
Training data samples	912,668	-	-
Evaluation data samples	228,167	60,000	288,167
Total data samples	1,140,835	60,000	288,167

**Table 6 sensors-23-08158-t006:** Accuracy results.

Metric/Dataset	Original	Generated	Combined
Accuracy	90.29%	92.09%	90.66%

## Data Availability

The data presented in this study are available on request from the corresponding author.

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
