# Peer review of "FoGGAN: Generating Realistic Parkinson’s Disease Freezing of Gait Data Using GANs"

_sensors, 2023, doi:10.3390/s23198158_

Round 1

Reviewer 1 Report

The main goal of this study was to present a GAN architecture for generating almost identical medical data for PD and specifically for FoG cases. The data which was used as input for the GAN deployed in this study was the ‘data_daphnet_combined’ dataset. Based on this kind of data inputs, specifically tabular data format, the FoGGAN architecture has been shown to be able to generate almost identical data to the real one dataset. this work can be accpted and pulished after adressing following queries/concders/revisions:

1. Clearly mention the contribution in introduction section.

2. What is the primary challenge addressed in this research paper regarding data in the healthcare domain?

3. How does the unavailability of quality data affect the development of state-of-the-art Artificial Intelligence technologies in healthcare?

4. What role do Generative Adversarial Networks (GANs) play in addressing data scarcity issues in healthcare?

5. Table 6. Accuracy results. Only accuracy is considered as metic. need to include other metrics as well e.g. precision, recall etc.

6. Table 5 provides a summary of the training and evaluation sample sizes for three distinct datasets employed in this study. How was this architecture decided? discuss.

7. Provide details of hyperparameter tuning.

8. How does the FoGGAN architecture contribute to data augmentation in the context of Parkinson's Disease (PD) research?

9. What metrics and methods were used to evaluate the similarity between the generated data and the original input data?

10. Why is it essential that the data generated by FoGGAN do not contain sensitive information?

Reviewer 2 Report

Title Clarity: The title should be more descriptive and concise. Consider something like "Data Augmentation for Parkinson's Disease using FoGGAN: A GAN-based Approach." Abstract Length: The abstract is quite lengthy. It should be condensed to include only the most essential information, focusing on the problem, methodology, and key results. Introduction Clarity: The introduction should provide more context about Parkinson's Disease, Freezing of Gait (FoG), and the specific challenges in obtaining FoG data. This will help readers understand the significance of your research from the beginning. Motivation Statement: Clearly state why generating synthetic FoG data is crucial. Explain how it addresses the data scarcity issue in Parkinson's Disease research and its implications for AI applications. GAN Definition: Provide a brief explanation of Generative Adversarial Networks (GANs) for readers who may not be familiar with the concept. Research Objectives: Clearly state the research objectives or questions your study aims to address in the introduction. Dataset Information: Elaborate on the FoG dataset used. Provide details about its source, size, and any preprocessing steps applied to it. Methodology Explanation: Explain the FoGGAN architecture and the GAN-based approach in more detail. Include information about the GAN's architecture, loss functions, and training process. Evaluation Metrics: Describe the specific metrics used to measure the similarity between generated and original data in more detail. Ethical Considerations: Discuss the ethical considerations of generating synthetic healthcare data, especially in the context of sensitive patient information. Results Presentation: Present the results of your experiments in a clear and organized manner. Use tables and figures to illustrate the key findings. Discussion of Results: Interpret the results and discuss their implications for the healthcare field and AI applications. Explain how the high-quality synthetic data can benefit AI models. Comparison with Related Work: Compare your approach to existing methods for data augmentation in healthcare and highlight the advantages of FoGGAN. Classifier Performance: Provide more details about the Deep Neural Network (DNN) classifier used, including its architecture and training process. Accuracy Results: Mention the specific accuracy values achieved by the DNN classifier for each of the three evaluation datasets. Limitations: Discuss the limitations of your approach, including any potential shortcomings or challenges encountered during the study. Future Work: Suggest possible directions for future research or improvements to FoGGAN. Citation Style: Ensure consistent and proper citation style throughout the manuscript, following a recognized academic style guide (e.g., APA, IEEE). Figures and Tables: Check the quality of figures and tables for clarity and readability. Ensure they are appropriately labeled and referenced in the text. Grammar and Language: Carefully proofread the manuscript for grammatical errors and language clarity. Abstract Conclusion: The abstract should conclude with a sentence summarizing the main takeaway or contribution of the study. Flow and Structure: Ensure a logical flow and structure in the manuscript, with clear transitions between sections. In-text Citations: Ensure that all claims and statements are properly supported by references to relevant literature. Conclusion: Summarize the key findings and contributions of your research in the conclusion section. References: Check the accuracy and completeness of the reference list. Ensure that all cited works are included. Clarity of Abbreviations: Spell out abbreviations when they are first introduced, e.g., "Generative Adversarial Networks (GANs)." Ethical Approval: Mention if ethical approval was obtained for any aspects of the research, especially when dealing with healthcare data. Data Privacy: Discuss how data privacy concerns were addressed when working with sensitive medical data. Reproducibility: Mention if the code and data used in the study are available for reproducibility. Abstract Revision: Revise the abstract to provide a clearer and more concise overview of the study's key contributions and findings.
